# Variability of Seed Germination and Dormancy Characteristics and Genetic Analysis of Latvian *Avena fatua* Populations

**DOI:** 10.3390/plants10020235

**Published:** 2021-01-26

**Authors:** Jevgenija Ņečajeva, Māra Bleidere, Zaiga Jansone, Agnese Gailīte, Dainis Ruņģis

**Affiliations:** 1Institute for Plant Protection Research ‘Agrihorts’, Latvia University of Science and Technology, LV-3004 Jelgava, Latvia; 2Stende Research Centre, Institute of Agricultural Resources and Economics, Dizzemes, Dizstende, Libagu Parish, LV-3258 Talsu District, Latvia; mara.bleidere@arei.lv (M.B.); zaiga.jansone@arei.lv (Z.J.); 3Latvian State Forest Research Institute ‘Silava’, LV-2169 Salaspils, Latvia; agnese.gailite@silava.lv (A.G.); dainis.rungis@silava.lv (D.R.)

**Keywords:** dormancy genotyping, gibberellin, iPBS markers, seed morphology, wild oat, weed

## Abstract

*Avena fatua* is an economically detrimental weed found in cereal growing areas worldwide. Knowledge about the variation of dormancy and germination characteristics, as well as of the genetic diversity and structure can provide additional information about different populations within a region. Identification and development of potential indicators or markers of seed germination behavior, would be useful for modelling emergence and seed bank dynamics. This study aimed to describe the initial germination, dormancy, and morphological characteristics of seeds from different Latvian populations of *A. fatua* and to investigate the genetic structure of local populations. Seed samples from 26 to 41 locations in different regions of Latvia were collected over three years. Seed morphology, initial germination, and the effect of dormancy-breaking treatments were determined. Seeds from 18 Latvian populations with contrasting seed dormancy characteristics were genotyped. Although morphological differences between seeds from different regions were revealed, genetic analysis of the selected populations indicated that the population structure was not related to geographical location, which probably reflects the impact of anthropogenic dispersal of *A. fatua* seeds. The effect of dormancy-breaking treatments varied between years, between climatic zones and between populations, indicating environmental influences as well as inherited dormancy.

## 1. Introduction

Wild oat (*Avena fatua* L.) is one of the most economically detrimental weed species in cereal growing areas worldwide [1]. It is a highly variable species and different local ecotypes have evolved as plants have adapted to local climatic conditions [2]. Seeds of *A. fatua* can also be transported over short or long distances with contaminated seed material and agricultural machinery [3]. As a result, genetically, morphologically, and physiologically different populations may exist in the same area. Reliable models of weed life cycle and soil seed bank dynamics are necessary for the decision support systems to design cropping systems with efficient weed control [4]. Models that predict seed dormancy loss in soil seed bank and germination with high precision have been developed for *A. fatua*, but the authors have cautioned that empirical data from different regions are necessary to adapt the models for use in these regions [5]. Seed dormancy is a trait with potentially high intraspecific variability. In *A. fatua*, variability can be manifested at several levels, from seeds within an individual plant to different populations [6] and is determined by genetic differences and the effect of the environment before and during seed maturation [7]. Answering the following questions would help to understand whether a common model can be used in a particular area or, if not, which of the several developed models should be applied in each case: (1) How different are the levels of dormancy between the populations in the area? (2) Are there regional or other patterns in the germination and dormancy behavior of the seeds? Additionally, morphological or other markers that indicate germination and dormancy behavior, either specific to a biotype or influenced by the environment of a particular growing season, would be useful for more extensive sampling and characterization of the populations.

Seed dormancy prevents germination of *A. fatua* directly after seeds are shed, as well as later in autumn, when germination is fatal in climates with winter temperatures falling below 0 °C. Seeds of *A. fatua* are mostly dormant at maturation, but the depth of dormancy depends on the genotype. It has been previously estimated that 50% of the dormancy phenotype is genetically determined and 50% is determined by the influence of the maternal environment [7]. Genetic models have been proposed to explain the heritability of the dormant seed phenotype in *A. fatua*, where dormancy is regulated by two germination-promoting loci and one dormancy-promoting locus [8].

Dormancy in *A. fatua* is lost in the process of dry afterripening and, gradually, the requirement for exogenous gibberellins that induce germination, is reduced [9]. The duration of afterripening that is required to completely relieve dormancy depends on the genotype and temperature during afterripening; the process is faster at higher temperatures [10]. It has been reported that the dormancy state was maintained in seeds stored at 5 °C [11], therefore, the minimum temperature required for the afterripening process is higher than 5 °C. A threshold value of gibberellic acid (GA) concentration required to induce germination or break dormancy is characteristic of particular populations [9]. Therefore, screening the response of seeds from different populations to the same concentration of GA could enable estimation of the degree of responsiveness of the seeds to GA and to compare dormancy levels of these seed populations.

Dormancy status of seeds with a similar genotype is also influenced by the parental environment. The effect of the parental environment on seed dormancy and germination is an important part of the ability of plants to sense the environment and adapt to it [12]. For example, seeds of *Amaranthus retroflexus* L. that have matured in different months have varying dormancy levels, characterized by the requirement for afterripening and the effect of light on germination [13]. In Northern climates, dry conditions and higher temperatures at certain critical stages of *A. fatua* seed maturation decrease dormancy, as well as seed mass and water content [10,14]. Shading of the mother plant, as well as drought, decreases both dormancy and resource allocation to the seeds [15,16]. Therefore, the germination behavior of one population can depend on the growth conditions of the particular year.

Germinability and afterripening requirement of *A. fatua* seeds are influenced by the seed hull. Dehulled seeds germinate both faster and to a higher percentage and the afterripening time is reduced [10,17]. However, dehulling and piercing of the dormant seeds is not sufficient for germination and seeds can remain dormant unless additional dormancy-breaking treatments are applied [18]. Drought and shading stress experienced by the mother plants decrease both seed and hull mass, as well as the content of phenolic compounds in the seeds and the hull [15]. At the same time, it is not clear whether seeds with relatively thicker hulls exhibit deeper dormancy than seeds with similar genotype and seed mass but with less developed seed hulls. The response of both these seed characteristics to stress can be variable between years even in genetically homogenous isolines of *A. fatua* [15].

Despite its economic importance, the genetic diversity and structure of *A. fatua* populations has not been extensively studied using molecular markers. Genetic differences between *A. fatua* populations in Pakistan were studied by analysing seed storage protein band polymorphism [19]. Two studies utilized inter simple sequence repeat (ISSR) and random amplified polymorphic DNA (RAPD) markers to investigate *A. fatua* populations in England [20] and the USA [21], both in the context of herbicide resistance. An additional study reported on the genetic diversity of *A. fatua* populations in China and the USA utilizing simple sequence repeat (SSR) or microsatellite markers [1]. The use of SSR markers for the determination of genetic diversity and population structure is advantageous, as they are co-dominant, informative, transferable, and repeatable. One disadvantage is the considerable effort involved in identifying the unique DNA sequences flanking the repeat regions to develop PCR primers. In many cases, SSR markers developed in one species can be transferred to related species. Cultivated oat (*A. sativa* L.) SSR markers as well as barley and wheat SSR markers have been tested and utilized for genotyping of *A. fatua* samples. However, the success rate in transferring SSR markers from related species to *A. fatua* was low, from 377 tested SSR markers, only 20 were utilizable for genotyping *A. fatua* [1]. An alternative is to use non-species-specific markers. Inter primer binding site (iPBS) markers have been developed based on retrotransposon long terminal repeat (LTR) sequences, which are ubiquitous in most plant and animal genomes [22]. These markers are informative in a wide range of species and are technically uncomplicated to implement in most molecular genetic laboratories [23]. Genetic diversity and population structure analysis results using iPBS markers are similar to those obtained by other anonymous marker methods such as ISSR [24,25].

In the 1940s and 1950s, *A. fatua* was widespread in the southern part of Latvia and occasionally found in a few other regions, and the results of a survey performed at the end of the 1990s were similar [26]. However, in the past two decades, wild oats have spread widely in arable lands throughout the entire territory of Latvia, as was shown by the monitoring of *A. fatua* distribution carried out in 2013–2017. There is a reason to suspect that the increased contamination was partly a result of imported contaminated seed material, while partly it was due to the use of local non-certified seed material.

Researchers from Poland and Lithuania have described several forms or varieties within the species that are characterized by seed morphological features, i.e., hull (lemma) and basal callus pubescence [27,28]. In Lithuania, the following four varieties have been described based on seed morphology: *A. fatua* var. *fatua* (syn. var. *pilosissima*), var. *intermedia*, var. *grabrata,* and var. *vilis*. Mostly one variety dominates in a plot of one hectare, with other species contributing one to five percent, if present. Within the 20 populations from Lithuania included in the analysis, the dominant variety was *A. fatua* var. *vilis* (66.5%), while var. *fatua* was found only in 1.6% [28]. In contrast, *A. fatua* var. *fatua* was found in 50.76% cases in the surveyed area in the south of Poland [27].

The aim of this study was to compare the morphological and physiological characteristics of seeds and to characterize the genetic structure of Latvian populations located in different regions of the country. Samples of wild oat seeds were collected in contaminated fields throughout Latvia in three consecutive years. Seed morphological characteristics, initial germination, and physiological characteristics related to dormancy (response to dry afterripening and treatment with exogenous GA) were compared between populations and years. Selected populations were genotyped using iPBS markers to determine genetic similarity of the populations and the variability of individuals within each population.

## 2. Results

### 2.1. Seed Germination and the Effect of Dormancy-Breaking Treatments

Average initial germination at 22 °C varied from 0 to 50% and was significantly different between populations and climatic zones, as well as between years in samples collected within the same population (Table 1, Figure 1a). The highest initial percentage of germinated seeds was reached in 2015 in populations Z3 (50%), and Z8 (40%). However, the initial germination percentage was mostly low (median value 5%).

Scarification increased the germination percentage of the seeds; the germination percentage of the scarified seeds varied from 23 to 91% (median value 75%). When testing the effect of the year and climatic zone on the germination of scarified seeds, all two-way interactions were significant (*p* < 0.0001) (Table 1). The mean germination percentage of scarified seeds was positively correlated with the initial germination percentage of untreated seeds (Pearson correlation coefficient 0.35, *n* = 41, *p* = 0.057) and the germination of afterripened seeds (Pearson correlation coefficient 0.30, *n* = 41, *p* = 0.024).

Germination of untreated seeds at 10 °C was higher as compared with germination at 22 °C (Figure 1a,c). The germination percentage at 10 °C ranged from 1% to 57% (median value 16%). There were significant interactions (*p* < 0.0001) between each of the factors in the regression model (population, zone, and year of collection) (Table 1).

There were significant interactions (*p* < 0.0001) between each of the factors in the regression models for the seeds germinated after each of the dormancy-breaking treatments (Table 1). The percentage of germinating seeds in the afterripening treatment varied from 8% to 98% (median 49%) (Figure 1b). The percentage of germinating seeds in the GA treatment varied from 14% to 100% (median 90%) (Figure 1d). Treatment with GA increased the proportion of germinating seeds by 6% to 93% as comparing with germination of untreated seeds at 10 °C (Figure 1c).

### 2.2. Varieties and Seed Morphological Characteristics

Despite the different number of samples analyzed each year, the relative proportion of each variety was consistent throughout the three-year period. The dominating variety was *A. fatua* var. *grabrata* followed by var. *fatua* and var. *vilis*, while var. *intermedia* was the least frequent variety (Table 2). Regional differences were also observed. In the Eastern region of Latgale, the proportion of the *fatua* variety was the highest, while in Zemgale (southern part of the West zone) it was low (Table 2). In 10 populations, one variety was consistently dominant over three years, i.e., *A. fatua* var. *fatua* (L1, L3, L9), var. *grabrata* (V3, V6, V7, Z1, Z5, Z10), and var. *vilis* (K10). In six populations where sampling was only done twice, in both years, one variety was dominant, i.e., var. *intermedia* (K2), var. *grabrata* (K9, Z4, Z6, V4), and var. *vilis* (K3). In 12 populations, one variety clearly dominated in two years, but in one year, the proportion within the population was different. In four populations, no variety dominated in any of the years (L2, L4, L8, and L10).

When comparing the initial germination between the varieties, germination percentage was significantly higher in var. *grabrata* than in var. *fatua* (*p* < 0.0001) and there was no significant interaction between the variety and year, although year was a significant factor (*p* < 0.0001). However, in the model with data from the afterripened seeds, there was a significant interaction between the variety and the year (*p* < 0.0001). In 2015, the germination percentage was lower in var. *fatua* (40%) than in var. *grabrata* (55%), while in 2016 and 2017 the results were opposite (72% and 34%, 58% and 45%, accordingly).

Mean mass of 100 seeds varied from 1.80 to 3.88 g between the accessions. There was a significant interaction between the year and the climatic zone (F = 3.14, DF = 4, *p* = 0.0158), as well as between the population and zone (F = 26.58, DF = 19, *p* < 0.0001), and population and year (F = 12.77, DF = 28, *p* < 0.00001). The values were lower in the West zone and higher in the Central and East zones (Figure 2a). Seed mass was significantly negatively correlated with the seed hull percentage in 2017 (Table 3).

The hull percentage varied from 29.97% to 45.5% (mean 37.1%, median value 36.9%). There was a statistically significant interaction of year and the climatic zone (F = 4.67, DF = 4, *p* = 0.0019). In the East zone, the hull percentage was lower in 2015 (34.7%) as compared with the West and Central zones (37.6–37.7%). There was no difference between the zones in 2016 (36.4–38.4%), while in 2017 the mean hull percentage was highest in the West zone (39.0%), while in the Central zone it was significantly lower (35.1%), and in the East zone there was no significant difference comparing to either of the other zones (36.8%) (Figure 2b).

There was no statistically significant correlation between the hull percentage and germination, except in 2017, when there was a significant negative correlation between the proportion of germinated seeds germinated at low temperature and those subjected to dormancy-breaking treatment and the hull percentage (Table 3). There was no correlation between the germination percentage of the scarified seeds and the hull percentage.

### 2.3. Genetic Differences between Populations

A total of 129 iPBS fragments were amplified from 358 *A. fatua* samples (Table 4, Appendix A). The number of fragments amplified by the seven utilized iPBS markers was similar in each population, ranging from 118 to 128 (average 123.55). However, the number of polymorphic fragments within each population was more variable, ranging from 56 to 105 (average 74.10). The proportion of polymorphic fragments within each population ranged from 0.43 to 0.81 (average 0.57). No private alleles, fragments uniquely amplified in one population, were found. Analysis of molecular variance (AMOVA) revealed that 77% of the total variation was attributed to genetic variation among individuals within populations, and 23% between populations (PhiPT = 0.226, *p* = 0.001).

Population pairwise PhiPT values ranged from 0.021 to 0.445, and the majority were significant (*p* < 0.01) (Table 5). Bayesian analysis was implemented in the STRUCTURE software, and the delta K method [30] identified K = 2 as the most likely number of clusters (Figure 3). This division into two clusters was also seen in the pairwise PhiPT values and the PCoA of Nei genetic distances between both populations and individuals (Figure 4). However, this clustering into two groups was not related to geographic location or other variables (e.g., dominant variety). Mantel analysis comparing the pairwise Nei genetic distances between the analyzed Latvian populations with the pairwise geographic distances did not reveal any significant correlation (R = 0.015, *p* = 0.380). For example, the four pairs of populations with the lowest (and non-significant) PhiPT values (K1-L6, V5-Z8, K10-Z8, and K10-V5) are not geographically close, and do not share other factors that were examined in this study. The genetic diversity (proportion of polymorphic loci) within populations was not associated with the number or proportion of *A. fatua* varieties present.

## 3. Discussion

The percentage of germinating seeds in freshly collected samples and dormancy levels differed both between populations and years. While in some of the populations (e.g., L1, L9, V9) the response to dormancy-breaking treatments was stable over the three years, in general, this response varied and the differences could not be explained by variations in hull percentage or seed mass in all years. However, the strong negative correlation of germination and seed hull proportion, in 2017, suggests that seed hull percentage can indicate potentially lower germinability. Hull thickness is a highly heritable characteristic in the primary kernels of cultivated oat varieties, but even in cultivated oats, the hull percentage varies significantly among the locations where oats are grown and depends on environmental conditions [31]. Drought and shading stress affect seed mass and hull mass differently, and this effect is also dependent on genotype [15]. Therefore, the resulting hull percentage is not a stable characteristic of a wild oat population, even if the population is genetically homogenous. More detailed research is required to understand the heritability of hull characteristics in *A. fatua* seeds and whether variations in the development of the hull can be predicted from the meteorological data in a particular season. More samples per population should be analyzed to ensure that the hull percentage of the germinated seeds is adequately assessed, because it is not possible to determine the germination percentage and hull percentage in the same seeds. Alternatively, non-destructive methods such as analysis of X-ray images of the seeds can be considered.

Seed hull proportion and pubescence can influence the dynamics of soil seed bank due to altered susceptibility of the seeds to pathogenic fungi. Seeds with more developed hulls were colonized by *Fusarium*, but not *Alternaria* or *Cladospodium* [32]. However, the hull may serve a protective function preventing the fungi from reaching the kernel [32]. To gain more information on the factors that influence the hull percentage, samples should be collected in specific sites where growth conditions are monitored, and the samples could be further separated into morphological varieties to test whether and how the hull percentage is related to other morphological and physiological characteristics of the seeds. Repeated sampling during one season could test the effect of growth conditions by distinguishing cohorts of plants where maturing seeds are exposed to different environmental conditions. Additionally, quantification of the total phenolic contents of the seed hull and the contents of particular compounds related to seed predation and resistance to fungal attack, and therefore potential persistence in soil, would be useful.

Although there was an apparent difference between the initial germination percentage of the variety *fatua* and that of the variety *grabrata*, this can be an effect of the specific climatic zone, because the seed samples where the *fatua* variety dominated were mostly collected in the East zone. More focused experiments with morphologically different seeds collected at the same site, or at geographically close sites, are required to show whether hull pubescence, which largely defines the variety, is a marker of germination behavior.

The results of the germination tests suggest that the germination percentage of seeds germinated at low temperature with and without GA can be used to estimate the level of dormancy in freshly collected seed samples. The correlation was not significant in 2015, probably because samples from the West climatic zone were not germinated at low temperature in 2015. Estimation of the cardinal temperature values of germination and the threshold temperature of afterripening in different populations, in relation to seed morphology, would be necessary to model germination and emergence rates of the seeds in different populations.

Although dry afterripening breaks physiological seed dormancy in many plant species, further research on the effect of cold stratification in *A. fatua* seeds is required. In northern climates, exposure to low positive temperatures is likely to be a major dormancy-breaking factor. The understanding of dormancy mechanisms is important to predict seed behavior with respect to climate change, because both germination and dormancy loss are influenced by temperature and other environmental factors [33]. In the case of *A. fatua* in northern climates, dormancy ensures that seeds remain in the soil seed bank throughout the winter when seedlings or plants cannot survive. However, as shown in this study, a proportion of seeds (in some cases, up to 50%) were non-dormant and able to germinate in autumn. If, due to climate change, winter conditions are modified so that seedlings can survive overwintering, early germination could become advantageous.

The morphological diversity of wild oat seeds makes it challenging to characterize each population. In cases when seeds with contrasting morphology were present in a sample, seeds of the dominating type were used for further analysis. The thousand seed mass was 22.2 g in a sample collected in Sweden [34]. The thousand seed mass was even smaller in *A. fatua* populations from Poland, i.e., 11.1–15.9 g [27]. In this study, only the basal (or primary) seeds from each spikelet were chosen, so the average mass would be lower if secondary seeds were taken into account. Our approach reduced the amount of information about seed germination in each population because the depth of dormancy is greater in the secondary seeds, which was confirmed by other studies [6,35] and our own observations (Necajeva, unpublished results). Although stress during seed maturation was reported to result in lower seed mass and reduced dormancy [15], there was an opposite correlation between seed mass and dormancy in 2016 and 2017 (Table 3). However, samples from different populations were used in the experiment described here, therefore, these results may not be comparable with the results reported by Gallaher and co-workers [15]. The seed mass was negatively correlated with the hull percentage, but it was not conclusively shown that the hull percentage is related to seed dormancy.

Screening of 34 oat, barley, and wheat SSR markers only identified four informative markers in *A. fatua*. This low transferability of SSR markers from oat, barley, and wheat to *A. fatua* has been reported previously [1]. Due to the small number of utilizable SSR markers, universal iPBS markers were used for genotyping. Given the allohexaploid genome of *A. fatua*, determination of allele dosage can be complicated, therefore, the use of dominant marker systems does not represent a large loss of information as compared with codominant marker systems (e.g., SSRs), particularly, if the number of loci able to be genotyped is larger (four SSR loci as compared with 129 iPBS loci in this study). Development or adaptation of additional markers will enable better resolution of the genetic relationships of *A. fatua* populations.

In general, the analyzed *A. fatua* populations were moderately to well differentiated. Bayesian analysis indicated that the analyzed populations were most likely divided into two clusters. However, in this study, it was not possible to determine the basis of this clustering. Mantel analysis revealed that there was no correspondence between the genetic distances between populations and geographic distance, and the clusters were also not related to the predominant variety in each population or other investigated factors. No information was available on possible anthropogenic transmission routes of *A. fatua* seeds between genetically similar populations, e.g., via farm machinery or the use of common seed sources. The Norwegian and Polish populations were assigned to different clusters, but within these clusters, they were not more highly differentiated between Latvian populations as compared with pairwise comparisons between populations within Latvia. This is consistent with reports documenting the long history of *A. fatua* presence in Latvian fields, representing repeated introductions over an extended time. Similar results were reported in a study of *A. fatua* seeds from different populations in Pakistan, where the variation of polymorphic protein bands was not related to the latitude and longitude of the collection sites, although some of the populations were well differentiated from others [19]. A high degree of genetic variability in *A. fatua* populations can be an adaptation to heterogenous environments [36]. In some populations, change in the varietal composition between years was observed, however, this study did not investigate changes in genetic parameters in populations over several years. Further investigations are required to determine if these observed changes in varietal composition are also reflected in changes in genetic parameters. However, no genetic clustering of populations with similar variety composition was observed. Another related aspect for further study is the stability of populations over time. Changes in populations could be due to supplementation by new germplasm from outside the population, or from soil seed banks that contain a diverse range of germplasm, with differential germination rates determined by environmental and other factors. In addition, decreased genetic differentiation has been described previously for populations of annual plant species with persistent soil seed banks [37,38].

The morphological characteristics of the seeds, as potentially useful genetic markers related to the germination characteristics, require more attention. Jain and Rai [39] distinguished two kinds of lemma hairiness (hairy vs. non-hairy) and two lemma color variations (black vs. grey) and reported a higher germination rate in seeds with black-colored lemma (hull). The effect of lemma color on seed germination also depends on the genotype and other characteristics that influence the depth of dormancy in *A. fatua* seeds [6]. In our study, the color of the hull varied from a yellow, light-, darkish-brown to a grey, darkish-grey (black), and there were three degrees of hull pubescence (i.e., lemma hairiness). Hull pubescence was used to determine the variety, but the hull color was not related to this feature. In further studies, seeds with different hull morphology collected at the same locations should be compared to be able to determine if the level of dormancy is related to the morphology.

In future research, to identify genetic markers related to the dormancy characteristics of the seeds, the same seeds that were germinated should be used for genotyping and morphological assessment. It is also important to determine whether there are differences between the types of plants or morphological varieties in terms of agronomically relevant characteristics, especially seed persistence in the soil seedbank. Comparing the prediction accuracy of existing models in small areas where plants with different seed characteristics dominate would be interesting.

## 4. Materials and Methods

Seed samples were collected in 41 sites in different parts of Latvia in 2015 and repeatedly collected in 26 of the original sites in 2016 and in 32 sites in 2017 (Figure 5). Seeds were collected from the end of July to mid-August, depending on when the seeds began to disperse in each of the years. We considered seeds collected in one field as belonging to one population of *A. fatua*. The number of plants sampled depended on the size of the colony, but at least 100 plants were sampled at each location. Most of the samples were collected in spring cereals (wheat, barley, or a mixture of cereals), but several samples were also collected in winter wheat, spring or winter oilseed rape, beans, peas, or fallow. Seeds were mostly collected in the fields with high density of *A. fatua*, but in some cases repeated collection was possible only in the field margins. Seeds were not collected if *A. fatua* was not present or the density of the plants was too low.

Latvia is situated in the Hemiboreal climate zone, the average elevation throughout the territory is 89.5 m a.s.l., ranging from 35 to 200 m a.s.l. in 97% of the territory [40]. Collection sites were situated in the following four regions of Latvia: Kurzeme, Latgale, Vidzeme, and Zemgale. Samples were coded accordingly (K1–K10, L1–L11, V1–V10, and Z1–Z10) and grouped by the climatic zones (West, East, and Central). The West zone has the least continental climate and in the Southwest zone thaws are frequent in winter. Climate continentality increases in the Central zone and is most pronounced in the East zone [29] (Figure 5). Samples collected in Kurzeme and Zemgale were assigned to the West zone (south and south-west of the country), except Z3, which belonged to the Central zone; Latgale samples and V10 from Vidzeme were assigned to the East zone; the other samples from Vidzeme were assigned to the Central zone (the central and northern region).

After collection, the seeds were dried at room temperature for 14 days. After drying, the samples were divided into two subsamples, one of which was stored at 5 °C and the other at room temperature (18–20 °C) for afterripening. In addition, *A. fatua* seeds from Poland and Norway were obtained for genetic analysis. One population from Poland was collected from arable fields in the Lesser Poland region in Kocmyrzów, collection date 7 July 2016, and one population from Norway was obtained from Akershus county, Ski municipality, collection date 8 August 2017. Coordinates of the collection sites are not given to protect the privacy of the field owners.

The initial germination was tested in the laboratory within 14 days after collection. Each year, after a period of afterripening (206–236 days, depending on the date of seed collection), seeds were germinated in the same conditions as when the initial germination was tested (22 °C). Additionally, the effect of low germination temperature and exogenous gibberellic acid was tested on seeds of at least populations from each zone (four from the East zone, four from the Central zone, and eight from the West zone), stored dry at 5 °C (seeds germinated at 10 °C with or without gibberellic acid). In 2015, seeds were also subjected to scarification treatment; each seed was punctured with a dissection needle at the dorsal side, opposite from the basal end of the seed. Mechanically scarified seeds were germinated at the same time and in the same conditions, as the seeds used for testing the initial germination.

All germination tests were performed in four replications of 25 seeds (individual caryopses with intact hull, except the scarification treatment). Seeds were surface sterilized by soaking in KMnO_4_ solution for 5 min and rinsed three times with pure water. The seeds were placed in plastic Petri dishes (diameter 9.0 cm) with two layers of filter paper, and another layer was placed above, and 5.0 mL of deionized water or gibberellic acid solution (GA_3_, 1.0 mM) was added. Petri dishes were placed in polyethylene bags to prevent moisture loss. Seeds were incubated at a constant temperature of 22 or 10 °C, in darkness, in an environmental test chamber (Sanyo, MLR-351H, Panasonic). Germinated seeds were counted and removed after three, six, ten, and 14 days from the beginning of the test and after that weekly for 42 days (if germination continued, up to 63 days). Exposure to light was not prevented when counting germinated seeds. The effect of GA and germination at low temperature was tested in 16 populations: K1, K5, K7, K10, L1, L3, L5, L9, V3, V5, V8, V9, Z3, Z4, Z5, and Z8. Seeds from Kurzeme and Zemgale were used for these tests only in 2016 and 2017, while seeds from other populations were tested in all three years of the study. Only the basal (larger) seeds were used for germination tests in this study, to reduce the variability in the germination test results.

Color of the seed hull, abundance of hairs on the hull (hull pubescence), and abundance of basal hairs around the scar area were characterized for each seed accession. In each sample, fifty seeds per accession were characterized and varieties (forms) of *A. fatua* were assigned according to the criteria suggested by Korniak and co-authors [28], i.e., hull and basal callus pubescence. The varieties *fatua* and *intermedia* have hairy hulls (more intense in *fatua*), while var. *grabrata* and var. *vilis* do not have pubescent hulls. The variety *vilis* is characterized by reduced scar (basal) pubescence. The variety was assigned to each seed and the percentage of seeds of each of the four varieties calculated for each sample; if more than 70% belonged to the same variety, the whole seed accession was assigned this variety, otherwise it was marked as mixed.

Average mass of 100 seeds was determined by weighing three replications of 100 seeds. Hull to seed mass ratio (hull percentage) was determined by weighing a sample of seeds (2.5–3.0 g), dehulling the seeds, and weighing separately the kernels. Hull to seed mass ratio (hull percentage) was calculated as percentage of the mass of the seed hulls in relation to the mass of the entire seeds. Hull to seed mass ratio and mass of 100 seeds were determined for 41 samples in 2015, for 16 samples in 2016, and 32 samples in 2015.

For DNA extraction, *A. fatua* seeds were germinated on filter paper soaked with deionized water in a growth chamber (16 h light at 22 °C and 8 h dark at 18 °C). DNA was extracted from seedling leaves using a modified CTAB method [41]. A total of 20 populations were analyzed, i.e., 18 Latvian populations, as well as one population from each of Poland and Norway.

A total of 12 *A. sativa* SSR markers were tested, i.e., AM2, AM3, AM4, AM11, AM14, AM17, AM21, AM22, AM23, AM25, AM38, and AM42 [42] on a panel of 15 randomly selected *A. fatua* DNA samples. Previous studies have utilized SSR markers developed for barley and wheat for genotyping of *A. fatua* [1]. Therefore, in addition to the oat SSR markers, 16 barley (*Hordeum vulgare*) SSR markers, i.e., Bmac0040, Bmac0134, Bmac0032, Bmag0125, Bmac0156, EBmac0701, Bmac0093, Bmag0353, Bmag0211, Bmac0399, HVM67, WMC1E8, Bmag0173, Bmag0135, Bmac0067, and Bmag0382 [43], and six wheat (*Triticum aestivum*) SSR markers, i.e., Xgwm294, Xgwm325, Xgwm111, Xgwm47, Xgwm131, and Xgwm219 [44], were tested. PCR primers were labelled with one of three fluorophores (6-FAM, HEX, or TMR), and PCR amplification products were visualized on an Applied Biosystems 3130xl Genetic Analyzer. Of these 34 SSR markers from *A. sativa* (12 markers), *H. vulgare* (18 markers), and *T. aestivum* (six markers) tested for use in *A. fatua*, only four *A. sativa* microsatellite markers were identified that were able to be unambiguously genotyped in a subset of *A. fatua* samples (AM2, AM4, AM14, and AM22). Therefore, a universal marker system (iPBS) was utilized for genotyping of the *A. fatua* populations. Genotyping was done with seven iPBS markers (2001, 2095, 2239, 2380, 2009, 2076, and 2220) [23] using 100 ng of DNA in a 25 μL PCR mixture containing 1x Dream Taq buffer, 2.5 mM MgCl2, 0.2 mM of each dNTP, 1 µM primer, 1 U Dream Taq polymerase, and 0.04 U Pfu polymerase. PCR amplification conditions were as follows: initial denaturation at 94 °C for 4 min, followed by 38 cycles of 94 °C for 20 sec, 50 °C for 1 min, and 68 °C for 1 min, final elongation at 72 °C for 5 min. Amplified products were separated and visualized using a LabChip GX Touch platform (Perkin Elmer, Waltham, MA, USA) with a 5K DNA chip, according to the manufacturer’s protocol. Amplified fragments were scored as binary genotypes (presence/absence).

The germination data were organized in the form of binary outcomes. Binomial logistic regression models were used to compare the proportion of germinated seeds between different populations, climatic zones, and between years, and two-way interactions between these three factors were included in the models. The initial germination, germination of afterripened seeds, and seeds treated with GA were analyzed separately. In the case of the seeds collected in 2015, an additional analysis was performed with the factors of population, climatic zone, and treatment (scarified vs. intact), with the two-way interactions between the factors. Additionally, to compare germination between the varieties, models where dominating variety, year, and the two-way interaction between these factors were used. The data subset included data from all seed accessions, but only the populations where the dominating variety was *grabrata* or *fatua*, because other varieties were less frequent. Analysis was performed separately on intact and afterripened seed germination data. Germination results for the seeds collected in 2015 from the populations that were used for genotyping were analyzed separately, because in many cases seeds could not be collected in the same populations in all three years. Goodness of fit of the models was assessed using the Hosmer–Lemeshow test, assuming that *p* > 0.05 indicates no significant lack of fit. The effect of climatic zones and morphological varieties on the mass of 100 seeds and the hull percentage of the seeds was determined using analysis of variance. Diagnostic plots were used to check the model assumptions. The statistical analysis was performed in R, version 4.0.1. [45].

Binary genotyping data were analyzed using GenAlEx 6.501 [46]. Pairwise Nei genetic distances were calculated between populations and individuals and visualized using PCoA. The partition of genetic diversity within and between populations was assessed by AMOVA using PhiPT (an analog of FST also used for dominant markers) with 9999 permutations. Markov chain Monte Carlo (MCMC) clustering of individuals into population groups was done using STRUCTURE 2.3.4 [47], assuming admixture and correlation of allele frequencies among populations. The number of burn-in MCMC repetitions was 10,000 and 20,000 after burn-in. The number of assumed groups (K) was set from 1–20. Inference of the most likely number of clusters was done using Structure Harvester [48] and CLUMPAK [49].

## 5. Conclusions

Germination and dormancy characteristics within most of the Latvian *A. fatua* populations vary between different populations, as well as between years. There is no correlation between genetic and geographic distance between *A. fatua* populations in Latvia, or with other factors investigated in this study. Genetic similarities between geographically distant populations suggest that seeds may be transported over longer distances with crop seed material or, possibly, farming equipment. The effects of environmental conditions and heritable components influencing the seed and seed hull morphology are potentially important for a better understanding of the dynamics of the seed population and soil seed bank of *A. fatua*.

## Figures and Tables

**Figure 1 plants-10-00235-f001:**
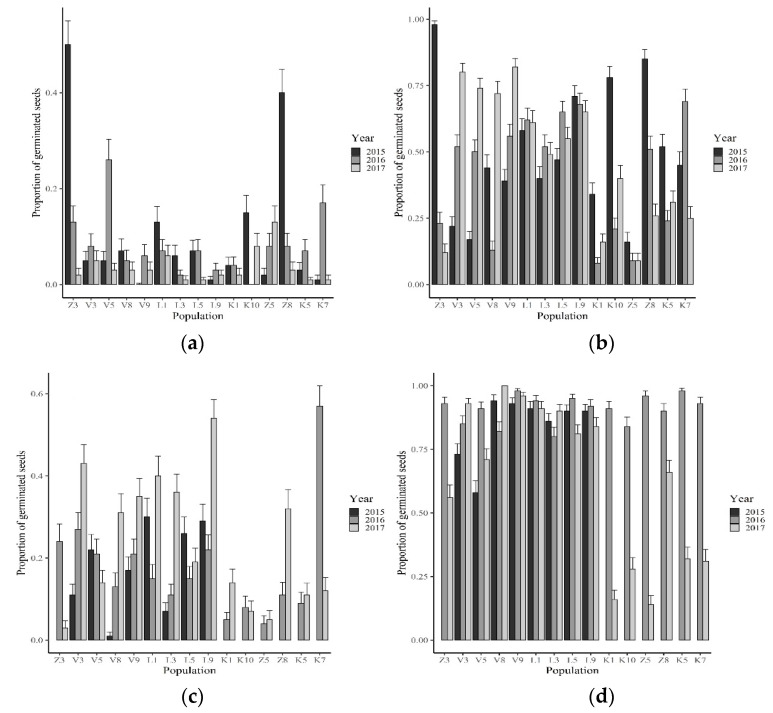
The proportion of germinated seeds of different Latvian *Avena fatua* populations in different germination and dormancy breaking treatments (proportion of the total number of seeds in the sample, vertical bars represent the standard error). (**a**) Initial germination tested within 2 weeks after seed collection; (**b**) Afterripening treatment (seeds germinated at 22 °C); (**c**) Untreated seeds (seeds germinated at 10 °C); (**d**) Treatment with 1.0 mM gibberellic acid. The presented data are back-transformed values of the logistic regression models.

**Figure 2 plants-10-00235-f002:**
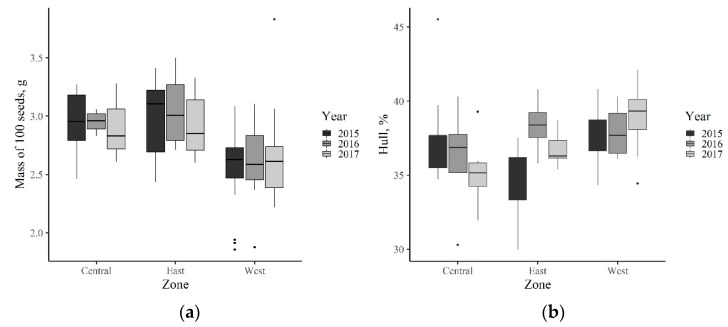
(**a**) Mass of 100 seeds and (**b**) hull percentage of seeds collected in populations located in different climatic zones in Latvia, in 2015–2017. Each box shows the interquantile range (25–75 percentile), the bars show the distance between the smallest and the largest point in the 1.5 times interquantile range, the dots are data points beyond this range.

**Figure 3 plants-10-00235-f003:**
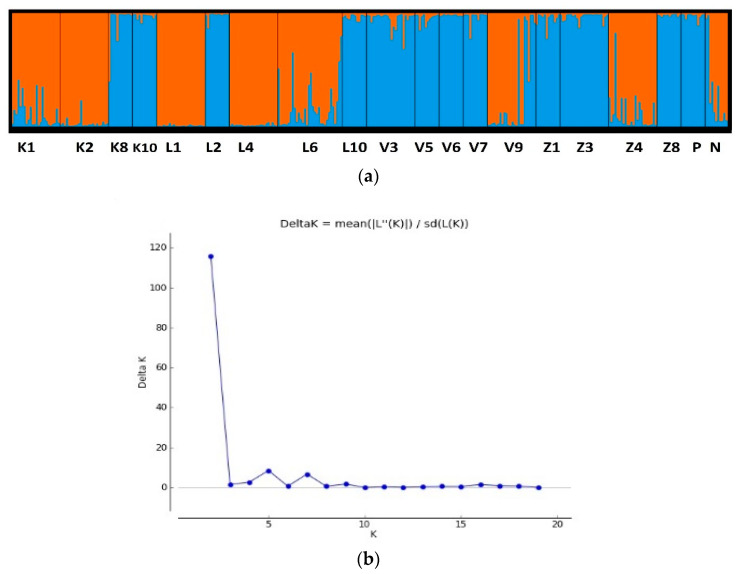
(**a**) Model-based analysis of iPBS genotypes assuming two subpopulations (K = 2) using the program STRUCTURE; (**b**) DeltaK analysis for 2z-19 clusters, indicating K = 2 as the most likely number of clusters.

**Figure 4 plants-10-00235-f004:**
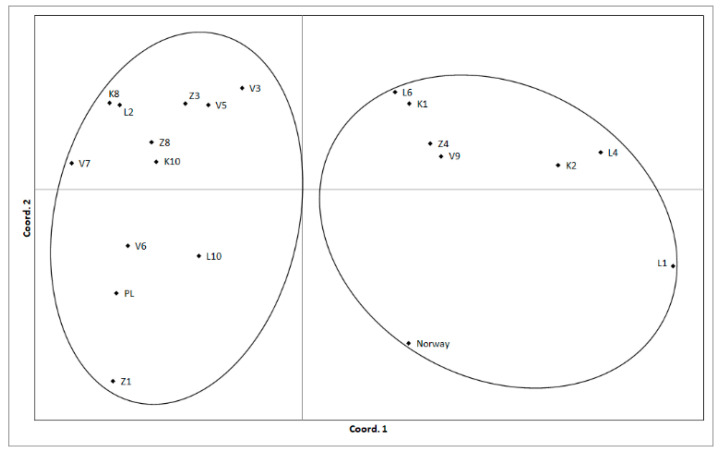
PCoA based on Nei genetic differences. Percentage of variation explained by the first 2 axes, 2.3 and 15.8, respectively. Clusters identified by STRUCTURE analysis are circled.

**Figure 5 plants-10-00235-f005:**
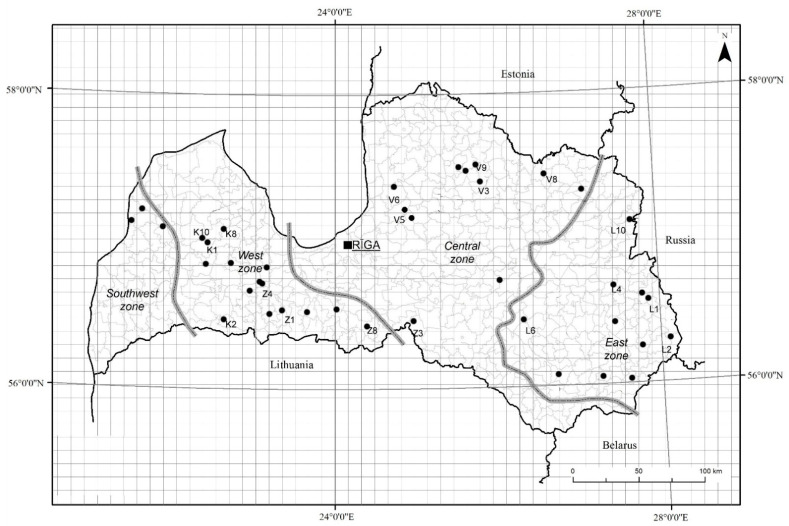
*Avena fatua* seed collection sites in Latvia. Labels added to the points showing sites where seeds used for genotyping were collected in 2015. Climatic zones are indicated according to the description in [29].

**Table 1 plants-10-00235-t001:** Deviance tables for the logistic regression models comparing germination of seeds from different populations and climatic zones, collected in three years, in different treatments. Initial, the initial germination tested within 2 weeks after seed collection; AR, afterripening treatment (seeds germinated at 22 °C); 10 °C, untreated seeds germinated at low temperature; GA, treatment with 1.0 mM gibberellic acid (germinated at 10 °C). Germination of intact and mechanically scarified seeds collected in 2015 and germinated at 22 °C within 2 weeks after collection was analyzed separately. Df, degrees of freedom; Resid. Df, residual degrees of freedom; Resid. Dev, residual deviance.

**Initial**	**Df**	**Deviance**	**Resid. Df**	**Resid. Dev**	***p***
	NA	NA	4499.00	2379.83	NA
Population	9	133.56	4490.00	2246.27	<0.0001
Zone	2	26.61	4488.00	2219.66	<0.0001
Year	2	63.59	4486.00	2156.07	<0.0001
Population:Zone	3	1.56	4483.00	2154.51	0.6680
Zone:Year	4	6.51	4479.00	2148.00	0.1639
Population:Year	18	191.90	4461.00	1956.09	<0.0001
**AR**	**Df**	**Deviance**	**Resid. Df**	**Resid. Dev**	***p***
	NA	NA	4489.00	6193.94	NA
Population	9	263.56	4480.00	5930.38	<0.0001
Zone	2	111.54	4478.00	5818.83	<0.0001
Year	2	22.49	4476.00	5796.34	<0.0001
Population:Zone	3	28.44	4473.00	5767.90	<0.0001
Zone:Year	4	174.50	4469.00	5593.40	<0.0001
Population:Year	18	518.04	4451.00	5075.35	<0.0001
**10 °C**	**Df**	**Deviance**	**Resid. Df**	**Resid. Dev**	***p***
	NA	NA	3787.00	3794.93	
Population	9	137.45	3778.00	3657.48	<0.0001
Zone	2	36.56	3776.00	3620.92	<0.0001
Year	2	39.50	3774.00	3581.42	<0.0001
Population:Zone	3	14.91	3771.00	3566.51	<0.0001
Zone:Year	3	43.69	3768.00	3522.82	<0.0001
Population:Year	13	161.81	3755.00	3361.01	<0.0001
**GA**	**Df**	**Deviance**	**Resid. Df**	**Resid. Dev**	***p***
	NA	NA	3797.00	3944.50	
Population	9	273.59	3788.00	3670.91	<0.0001
Zone	2	107.54	3786.00	3563.37	<0.0001
Year	2	397.06	3784.00	3166.31	<0.0001
Population:Zone	3	51.07	3781.00	3115.25	<0.0001
Zone:Year	3	178.04	3778.00	2937.20	<0.0001
Population:Year	13	136.63	3765.00	2800.57	<0.0001
**Scarification**	**Df**	**Deviance**	**Resid. Df**	**Resid. Dev**	***p***
			8199.00	10904.26	
Population	19	211.32	8180.00	10692.94	<0.0001
Zone	2	63.77	8178.00	10629.17	<0.0001
Treatment	1	4085.75	8177.00	6543.42	<0.0001
Population:Zone	19	210.17	8158.00	6333.26	<0.0001
Zone:Treatment	2	40.46	8156.00	6292.80	<0.0001
Population:Treatment	19	119.40	8137.00	6173.40	<0.0001

**Table 2 plants-10-00235-t002:** Proportions of varieties of *Avena fatua* (%) in the seed samples collected in 2015–2017. Number of characterized samples, in 2015, 41 samples; in 2016, 26 samples; in 2017, 32 samples. Seeds were collected in four Latvian regions, and a climatic zone was assigned to each of the collection sites using the description of climatic zones in Latvia [29].

Year	Region	Zone	% *Fatua*	% *Intermedia*	% *Grabrata*	% *Vilis*
2015	**Total**		18.3	14.2	48.7	18.7
	Latgale	East	44.5	5.5	31.3	18.7
	Vidzeme	Central	20.0	13.2	63.8	3.0
	Kurzeme	West	5.0	28.6	29.0	37.4
	Zemgale	West	1.0	10.6	72.6	15.8
2016	**Total**		15.4	6.6	58.2	19.8
	Latgale	East	53.0	1.0	33.0	13.0
	Vidzeme	Central	2.4	9.6	78.0	10.0
	Kurzeme	West	5.0	9.0	63.0	23.0
	Zemgale	West	5.7	6.3	51.7	36.3
2017	**Total**		19.1	10.3	49.9	20.7
	Latgale	East	57.7	7.3	14.3	20.7
	Vidzeme	Central	15.4	4.6	76.6	3.4
	Kurzeme	West	11.8	18.0	43.5	26.8
	Zemgale	West	2.3	11.8	49.8	36.3

**Table 3 plants-10-00235-t003:** Correlations between the mean mass of 100 seeds (Mass) and seed hull percentage (Hull), mean initial germination proportion (Initial), mean percentage of afterripened seeds (AR), seeds germinated at low temperature (10 °C), and seeds treated with 1.0 mM gibberellic acid (GA). In each section of the table, the Pearson correlation coefficients (lower left part) and corresponding *p* values (upper right part) are reported. Number of observations *n* = 15, except in 2015, where *n* = 8 for 10 °C and GA treatments.

2015	Mass	Hull	Initial	AR	10 °C	GA
Mass		0.4387	0.5892	0.7541	0.9427	0.7027
Hull	−0.2163		0.9331	0.7313	0.4136	0.3669
Initial	−0.1518	−0.0237		0.0010	0.8231	0.8961
AR	0.0884	−0.0969	0.7598		0.2738	0.0285
10 °C	0.0306	−0.3375	0.0949	0.4412		0.9411
GA	−0.1613	−0.3701	0.0555	0.7605	−0.0315	
2016	Mass	Hull	Initial	AR	10 °C	GA
Mass		0.2664	0.3919	0.0092	0.0342	0.5760
Hull	−0.3066		0.9132	0.3975	0.6084	0.3483
Initial	0.2385	−0.0308		0.4394	0.0461	0.3079
AR	0.6466	−0.2358	0.2160		0.0245	0.6472
10 °C	0.5487	−0.1441	0.5217	0.5764		0.7169
GA	−0.1571	−0.2605	0.2823	0.1288	0.1022	
2017	Mass	Hull	Initial	AR	10 °C	GA
Mass		0.0065	0.9397	0.0028	0.0375	0.0027
Hull	−0.6679		0.5824	0.0018	0.0363	0.0004
Initial	0.0214	0.1545		0.5480	0.4643	0.2501
AR	0.7147	−0.7360	−0.1686		0.0053	0.0001
10 °C	0.5406	−0.5434	−0.2047	0.6800		0.0006
GA	0.7158	−0.7910	−0.3167	0.8273	0.7778	

**Table 4 plants-10-00235-t004:** Number of loci, polymorphic loci, germination characteristics (proportion of germinating seeds, Initial (the initial germination), and AR (germination of afterripened seeds) tested at 22 °C and varietal composition in the analyzed *Avena fatua* populations. Multiple comparisons with Tukey correction were made between the populations within each treatment, different letters denote significant differences between the populations, *p* < 0.05).

Population	Number of Individuals Genotyped	Total Number of iPBS Loci	Number of Polymorhic Loci	Proportion of Polymorphic Loci	Average Unbiased He (SE)	*A. fatua* Varietal Composition in 2015	Initial	AR
K1	24	125	77	0.62	0.217 (0.018)	100% *vilis* (*grabrata* in other years)	0.02 abc	0.30 bc
K10	12	123	72	0.59	0.237 (0.020)	82% *vilis* (similar in other years)	0.15 c	0.78 fgh
K2	24	127	75	0.59	0.204 (0.018)	70% *intermedia* (similar in other years)	0.01 ab	0.90 hi
K8	12	120	79	0.66	0.239 (0.019)	90% *intermedia* (*grabrata* in other years)	0.00 a	0.55 de
L1	24	124	81	0.65	0.223 (0.018)	96% fatua (similar in other years)	0.15 c	0.62 ef
L10	12	121	61	0.50	0.180 (0.019)	40% *grabrata*, 26% *vilis*, 24% fatua	0.04 abc	0.57 def
L2	12	121	61	0.50	0.191 (0.020)	36% fatua 44% *grabrata*	0.07 abc	0.68 efg
L4	24	125	88	0.70	0.252 (0.018)	36% *intermedia* 36% *vilis* 24% fatua (similar in other years)	0.06 abc	0.64 ef
L6	32	125	78	0.62	0.213 (0.018)	98% fatua	0.11 bc	0.27 bc
V3	24	123	66	0.54	0.185 (0.019)	86% *grabrata* (similar in other years)	0.09 abc	0.27 bc
V5	12	124	67	0.54	0.208 (0.019)	98% fatua (*grabrata* in other years)	0.01 ab	0.01 a
V6	12	124	70	0.56	0.216 (0.020)	78% *grabrata* (similar in other years)	0.03 abc	0.36 cd
V7	12	118	56	0.47	0.172 (0.019)	100% *grabrata* (constant over 3 years)	0.03 abc	0.28 bc
V9	24	128	98	0.77	0.302 (0.018)	98% *grabrata* (changes in other years)	0.00 a	0.49 cde
Z1	12	124	66	0.53	0.192 (0.019)	90% *grabrata* (similar in other years)	0.01ab	0.27 bc
Z3	24	125	72	0.58	0.205 (0.019)	98% *grabrata* (*vilis* in other years)	0.50 d	0.98 i
Z4	24	124	105	0.85	0.304 (0.016)	96% *grabrata* (similar in other years)	0.00 a	0.14 b
Z8	12	123	60	0.49	0.179 (0.019)	98% *grabrata* (*vilis* in other years)	0.40 d	0.85 ghi
Poland	12	121	65	0.54	0.181 (0.018)	fatua		
Norway	12	126	85	0.67	0.283 (0.020)			

**Table 5 plants-10-00235-t005:** Population pairwise PhiPT values. PhiPT values below the diagonal, *p* values (9999 permutations) above the diagonal (only values of *p* > 0.01 shown).

	K1	K2	K8	K10	L1	L2	L4	L6	L10	V3	V5	V6	V7	V9	Z1	Z3	Z4	Z8	Poland	Norway
**K1**								0.017												
**K2**	0.219																			
**K8**	0.238	0.320																		
**K10**	0.245	0.284	0.120								0.027							0.109		
**L1**	0.280	0.103	0.401	0.359																
**L2**	0.243	0.320	0.108	0.072	0.401															
**L4**	0.172	0.070	0.294	0.279	0.149	0.309														
**L6**	0.021	0.224	0.234	0.232	0.294	0.232	0.188													
**L10**	0.330	0.316	0.239	0.164	0.353	0.195	0.324	0.321												
**V3**	0.254	0.293	0.158	0.136	0.377	0.124	0.290	0.243	0.286											
**V5**	0.209	0.239	0.100	0.036	0.343	0.074	0.234	0.189	0.179	0.092								0.145		
**V6**	0.239	0.280	0.108	0.081	0.357	0.097	0.292	0.234	0.154	0.210	0.073									
**V7**	0.245	0.361	0.067	0.178	0.445	0.158	0.316	0.244	0.302	0.258	0.164	0.082								
**V9**	0.140	0.132	0.176	0.129	0.179	0.193	0.051	0.147	0.219	0.183	0.106	0.188	0.218							
**Z1**	0.302	0.339	0.202	0.162	0.395	0.240	0.324	0.312	0.231	0.322	0.189	0.096	0.158	0.206						
**Z3**	0.276	0.290	0.157	0.118	0.382	0.139	0.292	0.260	0.227	0.088	0.064	0.164	0.225	0.190	0.278					
**Z4**	0.056	0.134	0.224	0.179	0.188	0.212	0.108	0.056	0.261	0.237	0.164	0.197	0.244	0.098	0.249	0.245				
**Z8**	0.252	0.293	0.168	0.021	0.395	0.119	0.302	0.217	0.242	0.193	0.021	0.079	0.213	0.169	0.221	0.145	0.186			
**Poland**	0.290	0.362	0.186	0.174	0.423	0.204	0.348	0.297	0.277	0.301	0.175	0.080	0.138	0.229	0.098	0.265	0.263	0.176		
**Norway**	0.260	0.191	0.290	0.206	0.235	0.283	0.205	0.271	0.311	0.302	0.225	0.204	0.304	0.142	0.229	0.281	0.186	0.255	0.282	

## Data Availability

The data presented in this study are available in Appendix A: Afatua_iPBS_genotypes.xlsx.

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
