# Peer review of "Variability of Seed Germination and Dormancy Characteristics and Genetic Analysis of Latvian Avena fatua Populations"

_plants, 2021, doi:10.3390/plants10020235_

Round 1

Reviewer 1 Report

Dear Authors,

The study presented in the manuscript No. 1068728 pt. 'Variability of seed germination and dormancy traits and genetic structure of the Latvian populations of Avena fatua' concerns the relationship between morphological, physiological and genetic traits of wild oat (Avena fatua L.) seeds coming mainly from different regions of Latvia. The presented research is of great practical importance, as the detailed knowledge of the germination biology of this plant may contribute to the development of cheap and ecological methods of eliminating this weed from agricultural crops. I believe that the research plan and methodology are adequate to the research goals set. The manuscript is well written but requires minor corrections or additions.

  1. correct the way of writing the cited literature in Reference section according to the requirements for the authors
  2. Line 91 write the full names of the abbreviations ISSR and RAPD
  3. Line 139 check if these data are really presented in fig. 2a and in table 2?
  4. Line 188 and 200 - where are the described data (table? fig?)
  5. Table 1- explain what the abbreviations used in the table mean
  6. Line 454 - shouldn't the mass of 1000 grains be determined?
  7. Line 460 specify what material the DNA was isolated from (seeds or seedlings?)
  8. Were the seeds surface sterilized for the germination test?

Author Response

Dear Reviewer, we are grateful for the useful comments and suggestions which, we believe, helped to improve the quality of this paper. Please find below our response to the comments, with the explanation of how the text was modified (marked in bold).

  1. correct the way of writing the cited literature in Reference section according to the requirements for the authors

The list of references was corrected.

  1. Line 91 write the full names of the abbreviations ISSR and RAPD

The full names were added in Line 91

  1. Line 139 check if these data are really presented in fig. 2a and in table 2?

The Table and figure numbers were corrected (fig. 1a and Table 1)

  1. Line 188 and 200 - where are the described data (table? fig?)

These data were not shown but only described in the text.

  1. Table 1- explain what the abbreviations used in the table mean

The abbreviations were explained (Df – degrees of freedom, Resid. Df. – residual degrees of freedom, Resid. Dev – residual deviance.)

  1. Line 454 - shouldn't the mass of 1000 grains be determined?

This is correct, however, the mass of 100 seeds is often used to determine the mass of 1000 seeds. We decided to report the original measurements, because the mass of 1000 seeds would be the same values multiplied by 10.

  1. Line 460 specify what material the DNA was isolated from (seeds or seedlings?)

The DNA was isolated from seedling leaves after the seeds were germinated (the information was added to the text)

  1. Were the seeds surface sterilized for the germination test?

Yes, the seeds were treated with KMnO4. (The description added to the text: “Seeds were surface-sterilized by soaking in KMnO4 solution (5 mg L-1) for 5 minutes and rinsed three times with pure water”)

Reviewer 2 Report

Title: Variability of seed germination and dormancy characteristics and genetic structure of Latvian Avena fatua populations can be improved as “Seed germination, dormancy and genetic diversity of Avena fatua populations from Latvia”; Abstract is concise but language needs to be revised by a native speaker; Introduction: the background and problem are clearly stated, but some more relevant papers can be added; Materials and Methods: some useful informations to geolocalize the study area are necessary to determine the ecological conditions in which the different populations of A. fatua live;  Results are generally acceptable, but no informations are reported about the varieties of A. fatua; Conclusions are in line by the obtained results, but may be improved; References are appropriate but could be updated. All the text require moderate English changes to improve the paper.

I have made a few suggestions in "Major Considerations". The following points may be considered while revising the manuscript:

- in lines 56 and 71, the authors give general information about seed dormancy which can also influenced by maternal/parental environment. I strongly suggest to add this following specific paper to improve the paper: https://doi.org/10.1080/11263504.2014.987845, where is reported that seed germination behaviour in common weed is not independent of the time of the year in which seeds are produced and is due to both the environmental conditions experienced by the mother plant during seed maturation and those experienced by seeds after seed set.

- in lines 44-51 to improve the references, I suggest to add this paper: https://doi.org/10.1002/ece3.5535, where the effect of environmental factors (temperature, light, storage time) on germination response and dormancy patterns in eight Mediterranean native wildplants were studied in terms of the dynamics of emergences and how this might be affected by climate changes

- In line 176 the authors report results about the varieties of A. fatua analysed, but in MM no informations are reported. Please add informations.

- In line 390 authors not give informations about the 41 sampling sites, i.e. Geographical coordinates, Altitude (m a.s.l.), Environmental characteristics (Climate, Light, Soil moisture), etc. These informations are necessary to determine the ecological conditions in which the different populations of A. fatua live.

- In line 435 authors not explain how the controls in dark conditions are carried out, please explains how the controls take place

Author Response

Dear Reviewer, we are grateful for the useful comments and suggestions which, we believe, helped to improve the quality of this paper. Please find below our response to the comments, with the explanation of how the text was modified (marked in bold).

Title: Variability of seed germination and dormancy characteristics and genetic structure of Latvian Avena fatua populations can be improved as “Seed germination, dormancy and genetic diversity of Avena fatua populations from Latvia”; Abstract is concise but language needs to be revised by a native speaker; Introduction: the background and problem are clearly stated, but some more relevant papers can be added; Materials and Methods: some useful informations to geolocalize the study area are necessary to determine the ecological conditions in which the different populations of A. fatua live;  Results are generally acceptable, but no informations are reported about the varieties of A. fatua; Conclusions are in line by the obtained results, but may be improved; References are appropriate but could be updated. All the text require moderate English changes to improve the paper.

We would like to propose an improved title: “Variability of seed germination and dormancy characteristics and genetic analysis of Latvian Avena fatua populations”

The text was revised and grammatical and spelling corrections were made.

We have included additional literature references and updated the Discussion (described in detail below).

I have made a few suggestions in "Major Considerations". The following points may be considered while revising the manuscript:

- in lines 56 and 71, the authors give general information about seed dormancy which can also influenced by maternal/parental environment. I strongly suggest to add this following specific paper to improve the paper: https://doi.org/10.1080/11263504.2014.987845, where is reported that seed germination behaviour in common weed is not independent of the time of the year in which seeds are produced and is due to both the environmental conditions experienced by the mother plant during seed maturation and those experienced by seeds after seed set.

A reference to the paper was added (Ln 56): “For example, seeds of Amaranthus retroflexus L. matured at different months have different dormancy levels, characterized by the requirement for afterripening and the effect of light on germination”

- in lines 44-51 to improve the references, I suggest to add this paper: https://doi.org/10.1002/ece3.5535, where the effect of environmental factors (temperature, light, storage time) on germination response and dormancy patterns in eight Mediterranean native wildplants were studied in terms of the dynamics of emergences and how this might be affected by climate changes

The paragraph in lines 44-51 refers specifically to Avena fatua, therefore we added the reference to the Discussion. We also speculated on the potential effect of the climate change: “Although dry afterripening breaks physiological seed dormancy in many plant species, further research on the effect of cold stratification in A. fatua seeds is required. In the northern climate, exposure to low positive temperatures is likely to be a major dormancy breaking factor. The understanding of dormancy mechanisms is important to predict seed behavior facing climate change, because both germination and dormancy loss are influenced by temperature and other environmental factors. In the case of A. fatua in the northern climate, dormancy ensures that seeds remain in the soil seed bank throughout the winter when seedlings or plants cannot survive. However, as shown in this study, a proportion of seeds (in some cases, up to 50%) were nondormant and able to germinate in autumn. If conditions in winter change so that the seedlings may overwinter, early germination could become advantageous.”

- In line 176 the authors report results about the varieties of A. fatua analysed, but in MM no informations are reported. Please add informations.

The term ‘varieties’ in this paper is used as in the paper by Korniak and co-workers, in the meaning of naturally occurring forms. This was explained in the Materials and Methods section:

 “In each sample, fifty seeds per accession were characterized and varieties (forms) of A. fatua were assigned according to the criteria suggested by Korniak and co-authors [28]: hull and basal callus pubescence. The varieties fatua and intermedia have hairy hulls (more intense in fatua), while grabrata and vilis don’t have pubescent hulls. The variety vilis is characterized by reduced scar (basal) pubescence. The variety was assigned to each seed and the percentage of seeds of each of the four varieties calculated for each sample, if more than 70% belonged to the same variety, the whole seed accession was assigned this variety, otherwise it was marked as mixed.”

- In line 390 authors not give informations about the 41 sampling sites, i.e. Geographical coordinates, Altitude (m a.s.l.), Environmental characteristics (Climate, Light, Soil moisture), etc. These informations are necessary to determine the ecological conditions in which the different populations of A. fatua live.

The location of the seed collection sites is shown in Figure 5. An additional reference to the figure was added to the text. In our opinion, it would not be acceptable to report the exact coordinates because this information may be sensitive to the owners of the fields.

All the collection sites were in the open crop fields, but no measurements of soil moisture or light levels at the sites were performed.

The following information was added to the text: “Latvia is situated in the Hemiboreal climate zone, the average elevation throughout the territory is 89.5 m a.s.l., ranging from 35 to 200 m a.s.l. in 97% of the territory.”

In addition, information about the climate zones was added: “The West zone has the least continental climate and in the Southwest zone thaws are frequent in Winter. Climate continentality increases in the Central zone and is most pronounced in the East zone.”

- In line 435 authors not explain how the controls in dark conditions are carried out, please explains how the controls take place

Although the seeds were incubated in darkness, the counting was performed under ambient light and no attempt was made to entirely prevent seed exposure to light. The explanation was added to the text: “Exposure to light was not prevented when counting germinated seeds.”

Reviewer 3 Report

The manuscript of Ņečajeva and colleagues is interesting and shows important ecological and genetic implications.

The authors, starting from several seeds of Avena fatua, collected in different areas of Lithuania, have measured polymorphisms and genetic heterogeneities correlating them to the germination processes.

The manuscript is clear, the initial hypotheses are supported by a considerable bibliography, and the results achieved are commented on and well discussed.

However, the authors should clarify a few critical issues:

  • The authors perform a genotyping of different seeds of Avena fatua. Furthermore, in the materials and methods they write: “In addition, 16 barley (Hordeum vulgare) SSR markers; Bmac0040,467 Xgwm325, Xgwm111, 470 Xgwm47, Xgwm131, Xgwm219 [40] were tested ”Why do they also conduct this analysis? They should better explain this.

  • In the text there are some typos such as p. 14, line 407

  • Finally, I suggest to the authors an interesting manuscript that discusses some mechanisms associated with germination:

Kumar A et al. Plant behaviour: an evolutionary response to the environment? Plant Biol (Stuttg). 2020 Nov;22(6):961-970. doi: 10.1111/plb.13149.

Author Response

Dear Reviewer, we are grateful for the useful comments and suggestions which, we believe, helped to improve the quality of this paper. Please find below our response to the comments, with the explanation of how the text was modified (marked in bold).

 The authors perform a genotyping of different seeds of Avena fatua. Furthermore, in the materials and methods they write: “In addition, 16 barley (Hordeum vulgare) SSR markers; Bmac0040,467 Xgwm325, Xgwm111, 470 Xgwm47, Xgwm131, Xgwm219 [40] were tested ”Why do they also conduct this analysis? They should better explain this.

The materials and method section was modified – ‘Previous studies have utilized SSR markers developed for barley and wheat for genotyping of A. fatua [1]. Therefore, in addition to the oat SSR markers, 16 barley (Hordeum vulgare) SSR markers……, and six wheat (Triticum aestivum) SSR markers……. were tested.’

The use of SSR markers from other species was mentioned in the introduction ‘Cultivated oat (A. sativa L.) SSR markers as well as barley and wheat SSR markers were tested and utilized for genotyping of A. fatua samples. However, the success rate in transferring SSR markers from related species to A. fatua is low, from 377 tested SSR markers, only 20 were utilisable for genotyping A. fatua [1].’

We believe that the inclusion of this information can be useful to further researchers, as it indicates the low rate of successful transfer of SSR markers from other species.

 In the text there are some typos such as p. 14, line 407

The typos in the text were corrected

 Finally, I suggest to the authors an interesting manuscript that discusses some mechanisms associated with germination:

 Kumar A et al. Plant behaviour: an evolutionary response to the environment? Plant Biol (Stuttg). 2020 Nov;22(6):961-970. doi: 10.1111/plb.13149.

We have added a reference to the paper: “The effect of the parental environment on seed dormancy and germination is an important part of the ability of plants to sense the environment and adapt to it”

Round 2

Reviewer 2 Report

The authors have provided the suggestions to improve the quality of the manuscript. The revised version may be considered for the pubblication in Plants journal